# FEDERATED LEARNING USING A MIXTURE OF EXPERTS

## ABSTRACT

Federated learning has received attention for its efficiency and privacy benefits, in settings where data is distributed among devices. Although federated learning shows significant promise as a key approach when data cannot be shared or centralized, current incarnations show limited privacy properties and have shortcomings when applied to common real-world scenarios. One such scenario is heterogeneous data among devices, where data may come from different generating distributions. In this paper, we propose a federated learning framework using a mixture of experts to balance the specialist nature of a locally trained model with the generalist knowledge of a global model in a federated learning setting. Our results show that the mixture of experts model is better suited as a personalized model for devices when data is heterogeneous, outperforming both global and local models. Furthermore, our framework gives strict privacy guarantees, which allows clients to select parts of their data that may be excluded from the federation. The evaluation shows that the proposed solution is robust to the setting where some users require a strict privacy setting and do not disclose their models to a central server at all, opting out from the federation partially or entirely. The proposed framework is general enough to include any kinds of machine learning models, and can even use combinations of different kinds.

## 1 INTRODUCTION

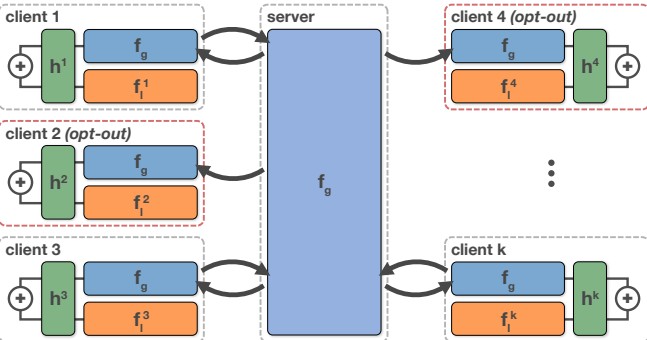

Figure 1: Overview: Federated mixtures of experts using local gating functions. Some clients opt-out from federation, not contributing to the global model and keeping their data completely private.

In many real-world scenarios, data is distributed over a large number of devices, due to privacy concerns or communication limitations. Federated learning is a framework that can leverage this data in a distributed learning setup. This allows for exploiting both the compute power of all participating clients, and to benefit from a large joint training data set. Furthermore, this is beneficial for privacy and data security. For example, in keyboard prediction for smartphones, thousands or even millions of users produce keyboard input that can be leveraged as training data. The training can ensue directly on the devices, doing away with the need for costly data transfer, storage, and immense compute on a central server (Hard et al., 2018). The medical field is another example area where data is extremely sensitive and may have to stay on premise, and a setting where analysis may require distributed and privacy-protecting approaches. In settings with such firm privacy

requirements, standard federated learning approaches may not be enough to guarantee the needed privacy.

The optimization problem that we solve in a federated learning setting is

$$\min_{w \in \mathbb{R}^d} \mathcal{L}(w) = \min_{w \in \mathbb{R}^d} \frac{1}{n} \sum_{k=1}^{n} \mathbb{E}_{(x,y) \sim p_k} \left[ \ell_k(w; \ x, y) \right] \tag{1}$$

where $\ell_k$ is the loss for client $k$ and $(x, y)$ samples from the $k$th client's data distribution $p_k$. A central server is coordinating training between the $K$ local clients. The most prevalent algorithm for solving this optimization is the federated averaging (FEDAVG) algorithm (McMahan et al., 2017). In this solution, each client has its own client model, parameterized by $w^k$ which is trained on a local dataset for $E$ local epochs. When all clients have completed the training, their weights are sent to the central server where they are aggregated into a global model, parameterized by $w^g$. In FEDAVG, the $k$ client models are combined via layer-wise averaging of parameters, weighted by the size of their respective local datasets:

$$w_{t+1}^g \leftarrow \sum_k \frac{n_k}{n} w_{t+1}^k, \tag{2}$$

where $n_k$ is the size of the dataset of client $k$ and $n = \sum_k n_k$. Finally, the new global model is sent out to each client, where it constitutes the starting point for the next round of (local) training. This process is repeated for a defined number of global communication rounds.

The averaging of local models in parameter space generally works but requires some care to be taken in order to ensure convergence. McMahan et al. (2017) showed that all local models need to be initialized with the same random seed for FEDAVG to work. Extended phases of local training between communication rounds can similarly break training, indicating that the individual client models will over time diverge towards different local minima in the loss landscape. Similarly, different distributions between client datasets will also lead to divergence of client models (McMahan et al., 2017).

Depending on the use case, however, the existence of local datasets and the option to train models locally can be advantageous: specialized local models, optimized for the data distribution at hand may yield higher performance in the local context than a single global model. Keyboard prediction, for example, based on a global model may represent a good approximation of the population average, but could provide a better experience at the hands of a user when biased towards their individual writing style and word choices. A natural question arises: when is a global FL-trained model better than a specialized local model? A specialist would be expected to perform better than a global generalist in a pathological non-iid setting, whereas the global generalist would be expected to perform better in an iid setting.

To address the issue of specialized local models within the federated learning setting, we propose a general framework based on mixtures of experts of a local and a global model on each client. Local expert models on each client are trained in parallel to the global model, followed by training local gating functions $h^k(x)$ that aggregate the two models' output depending on the input. We show advantages of this approach over fine-tuning the global model on local data in a variety of settings, and analyze the effect that different levels of variation between the local data distributions have on performance.

While standard federated learning already shows some privacy enhancing properties, it has been shown that in some settings, properties of the client and of the training data may be reconstructed from the weights communicated to the server (Wang et al., 2019). To this end, in this paper we will work with a stronger notion of privacy. While existing solutions may be private enough for some settings, we will assume that a client that require privacy for some of its data, needs this data to not influence the training of the global model at all. Instead, our framework allows for complete opting out from the federation with all or some of the data at any given client. Clients with such preferences will still benefit from the global model and retain a high level of performance on their own, skewed data distribution. This is important when local datasets are particularly sensitive, as may be the case in medical applications. Our experimental evaluation demonstrate the robustness of our learning framework with different levels of skewness in the data, and under varying fractions of opt-out clients.

## 2 RELATED WORK

Distributed machine learning has been studied as a strategy to allow for training data to remain with the clients, giving it some aspects of privacy, while leveraging the power of learning from bigger data and compute (Konečný et al., 2016; Shokri & Shmatikov, 2015; McMahan et al., 2017; Vanhaese-brouck et al., 2016; Bellet et al., 2018). The federated averaging technique (McMahan et al., 2017) has been influential and demonstrated that layer-wise averaging of the weights in neural network models trained separately at the clients is successful in many settings, producing a federated model that demonstrates some ability to generalize from limited subsets of data at the clients. However, it has been shown that federated averaging struggles when data is not independent and identically distributed among the clients (the non-iid setting), which shows that there is a need for personalization within federated learning (Kairouz et al., 2019).

In general, addressing class imbalance with deep learning is still a relatively understudied problem (Johnson & Khoshgoftaar, 2019). A common approach for personalization is to first train a generalist model and then fine-tune it using more specific data. This approach is used in meta-learning (Finn et al., 2017), domain adaptation (Mansour et al., 2009), and transfer learning (Oquab et al., 2014). This approach was proposed for the distributed setting by Wang et al. (2019) who used federated averaging to obtain a generalist model which was later fine-tuned locally on each client, using its specific training data. Some work has been inspired by the meta-learning paradigm to learn models that are specialized at the clients (Jiang et al., 2019; Fallah et al., 2020). Arivazhagan et al. (2019) combined this strategy and ideas from transfer learning with deep neural networks and presented a solution where shallow layers are frozen, and the deeper layers are retrained at every client.

Zhao et al. (2018) propose a strategy to improve training on non-iid client data by creating a subset of data which is globally shared between all clients. Recent strategies have also explored knowledge distillation techniques for federated learning (Jeong et al., 2018; He et al., 2020; Lin et al., 2020), which show promising results in non-iid settings.

Hanzely & Richtárik (2020) proposed a solution that provides an explicit trade-off between global and local models by the introduction of an alternative learning scheme that does not take the full federation step at every round, but instead takes a step in the direction towards the federated average. Deng et al. (2020) proposed to combine a global model $w$ trained using federated averaging, with a local model $v$ with a weight $\alpha_i$. To find optimal $\alpha_i$ they optimize $\alpha_i^* = \arg\min_{\alpha_i \in [0,1]} f_i\left(\alpha_i \boldsymbol{v} + (1-\alpha_i)\boldsymbol{w}\right)$ every communication round. While this weighting scheme will balance the two models, it has no way of adapting to the strengths of the different members of the mix.

Mixture of experts (Jacobs et al., 1991) is the combination of several competing neural networks trained together with a gating network to solve a common task. It was presented as an ensemble method which can be trained end to end using gradient descent. In the current work, we will apply the mixture to leverage the specific strengths of a global model trained with federated averaging, and a local model trained locally on each client.

## 3 FEDERATED LEARNING USING A MIXTURE OF EXPERTS

In this work, we present a framework for federated learning that builds on federated averaging and mixtures of experts. Our framework includes a personalized model for each client, which is included in a mixture together with a globally trained model using federated learning. The local models never leave the clients, which gives strong privacy properties, while the global model is trained using federated averaging and leverages larger compute and data. In our framework as seen in Figure 1, some clients can choose to opt-out from the federation, meaning that no information of their data is leaving the client, ensuring privacy of those clients.

Let $f_g$ be the global model with parameters $w_g$. We denote the index of clients by $k$ and the local models by $f_l^k$ with parameters $w_l^k$. The gating function is denoted by $h^k$, parameterized with $w_h^k$. Training in the proposed framework is divided into three main parts. First, a global model $f_g$ is trained using federated averaging using opt-in data (see Section 3.1). Second, a local model $f_l^k$ is trained using all available data on a client. Third, $f_g$ and $f_l^k$ are further trained together with a gating

model $h^k$ on each client locally, using all available data on the client. Steps one and two may be performed in parallel if allowed by the available resources.

### 3.1 Privacy guarantees

The proposed framework allows for a strict form of privacy guarantee. Each client may choose an arbitrary part of their data which they consider being too sensitive to use for federated learning, and no information from this data will ever leave the client. The system will still leverage learning from this data by using it to train the local model $f_l^k$ and the gating model $h^k$. This is a very flexible and useful property. For example, it allows for a user to use the sensitive data in training of the private local models, while transforming it using some privatization mechanism and use the censored version to train the federated model.

In general, each client dataset $\mathcal{D}^k$ is split into two non-overlapping datasets, $\mathcal{D}_\mathcal{O}^k$ and $\mathcal{D}_\mathcal{I}^k$, one of which has to be non-empty. The local model $f_l^k$ and the gating model $h^k$ is trained using the whole dataset $\mathcal{D}^k = \mathcal{D}_\mathcal{O}^k \cup \mathcal{D}_\mathcal{I}^k$, while the global model $f_g$ is trained with FEDAVG using only the *opt-in* dataset $\mathcal{D}_\mathcal{I}^k$. This is visualized in Figure 1.

### 3.2 Optimization

**Step 1: Federated averaging.** We train the global model using FEDAVG. In other words, globally we optimize

$$\min_{w_g \in \mathbb{R}^d} \mathcal{L}_{\text{global}}(w_g) = \min_{w_g \in \mathbb{R}^d} \frac{1}{|\mathcal{D}_\mathcal{I}^k|} \sum_{k \in \mathcal{D}_\mathcal{I}^k} \mathbb{E}_{(x,y) \sim \mathcal{D}_\mathcal{I}^k} \left[ \ell_k(w_g;\ x, y, \hat{y}_g) \right] \tag{3}$$

for the opt-in dataset $\mathcal{D}_\mathcal{I}^k$. Here $\ell_k$ is the loss for the global model $w_g$ on client $k$ for the prediction $f_g(x) = \hat{y}_g$, and $\mathcal{D}_\mathcal{I}^k$ is the $k$th clients *opt-in* data distribution.

**Step 2: Train local models.** The local models $f_l^k$ are trained only locally, sharing no information between clients, minimizing the the local loss over $w_l^k \in \mathbb{R}^d$,

$$\min_{w_l^k \in \mathbb{R}^d} \mathcal{L}(w_l^k) = \min_{w_l^k \in \mathbb{R}^d} \mathbb{E}_{(x,y) \sim \mathcal{D}^k} \left[ \ell_k(w_l^k;\ x, y, \hat{y}_l) \right] \quad \forall k = 1, \ldots, n. \tag{4}$$

Here $\ell_k$ is the loss for the prediction $\hat{y}_l = f_l^k(w_l^k;\ x)$ from the local model on the input $x$ and $\mathcal{D}^k$ is the $k$th clients dataset.

**Step 3: Train local mixtures.** The local mixture of experts are trained using the gating models $h^k$, with the prediction error given by weighing the trained models $f_g$ and $f_l^k$:

$$\hat{y}_h = h^k(x) f_l^k(x) + \left(1 - h^k(x)\right) f_g(x) \quad \forall k = 1, \ldots, n. \tag{5}$$

In other words, at the end of a communication round, given $f_l^k$ and $f_g$, we optimize the mixture equation 5:

$$\min_{w_g, w_l^k, w_h^k} \mathcal{L}(w_g, w_l^k, w_h^k) = \min_{w_g, w_l^k, w_h^k} \mathbb{E}_{(x,y) \sim \mathcal{D}^k} \left[ \ell_k(w_g, w_l^k, w_h^k;\ x, y, \hat{y}_h) \right], \tag{6}$$

locally for every client $k = 1, \ldots, n$. Here $\ell_k$ is the loss from predicting $\hat{y}$ for the label $y$ given the input $x$ with the model from equation 5 over the data distribution $\mathcal{D}^k$ of client $k$. A summary of the method is described in Algorithm 1.

---

**Algorithm 1**

---

1: **input:** Models participating in FEDAVG $w_1, \ldots, w_k$, local expert models $w_l^k$, local gate $w_h^k$, learning rate $lr$, decay rates $\beta_1, \beta_2$
2: Randomly initialize $w_1, \ldots, w_k$ with the same seed.
3: Randomly initialize $w_l^k$ and $w_h^k$.
4: $w_g \leftarrow$ FEDAVG$(w_1, \ldots, w_k)$ *// Train for E local epochs and G communication rounds*
5: **for** client $k$ **do**
6:     $w_l^k \leftarrow$ Adam$(w_l^k, lr, \beta_1, \beta_2)$ *// Train local model on client k*
7:     $w_g, w_l^k, w_h^k \leftarrow$ Adam$(w_g, w_l^k, w_h^k, lr, \beta_1, \beta_2)$ *// Train mixture of experts on client k*
8: **end for**

---

### 3.3 Experimental setup

**Datasets.** Our experiments are carried out on the datasets CIFAR-10, CIFAR-100 (Krizhevsky et al., 2009) and Fashion-MNIST (Xiao et al., 2017). In order to simulate heterogeneous client data *without labels overlapping between clients*, we partition the data into 5 clients for CIFAR-10 and Fashion-MNIST, and 50 clients for CIFAR-100. For CIFAR-100 we used a client sampling strategy where we in each communication round in the federation randomly sampled a fraction of 0.1 of clients to participate.

**Sampling non-iid.** The datasets are sampled in such a way that each client's datasets contains two majority classes which together form $p\%$ of the client data and the remaining classes form $(1 - p)\%$ of the client data. We perform experiments where we vary $p$ to see what effect the degree of heterogeneity has on performance. In the extreme case $p = 1.0$, each client only has two labels in total. *Note that this is an extreme partitioning, where there is no overlap of majority class labels between clients, i.e. the local data distributions $\mathcal{P}_i \cap \mathcal{P}_j = \emptyset$ for all pairs of clients $i, j$.* Also note that, we perform experiments with very little data in our experimental setup. A summary of the experimental set-up can be seen in Table 1.

**Opt-out factor.** Some users might want to opt-out from participating to a global model, due to privacy reasons. These users will still receive the global model. To simulate this scenario in the experimental evaluation, we introduce an *opt-out factor* denoted by $q$. This is a fraction deciding the number of clients participating in the FEDAVG optimization. The clients that participate in the federated learning optimization have all their data in $\mathcal{D}_{\mathcal{I}}^k$, while the rest of the clients that opt-out have all their data in $\mathcal{D}_{\mathcal{O}}^k$. $q = 0$ means all clients are opt-in and participating. We perform experiments varying $q$, to see how robust our algorithm is to different levels of client participation. In Figure 1 we visualize how the opt-out factor can be used.

**Models.** In our setup, both the local model $f_l$ and the global model $f_g$ are CNNs with the same architecture. However, they are not constrained to be the same model and could be implemented any two differentiable models. The CNN has two convolutional layers with a kernel size of 5, and two fully-connected layers. All layers have ReLU activations. The gating function $h_k$ has the same architecture as $f_g$ and $f_l$, but with a sigmoid activation in the last layer. We use Adam (Kingma & Ba, 2014) to optimize all models, with a learning rate of 0.0001 in all experiments.

**Baselines.** We use three different models as baselines. First, the locally trained model $f_l^k$ for each client. Second, FEDAVG. Third, the final model output from FEDAVG fine-tuned for each client on its own local data. We train $f_l^k$, the fine-tuned model and the mixture using early stopping for 200 epochs, monitoring validation loss on each client.

**Evaluation**. For evaluation we have a held-out validation set for each client. For both CIFAR-10 and CIFAR-100 we have $n = 400$ data points for evaluation per client, sampled with the same majority class fraction $p$. We report an average accuracy over all clients.

| Experimental set-up | |
|---|---|
| Datasets | CIFAR-10, CIFAR-100, Fashion-MNIST |
| Models $(f_g, f_l^k, h^k)$ | CNNs: 2 conv. layers (5x5 kernels) and two FC layers |
| Activation functions | ReLU |
| Communication rounds (FedAvg) | 1500 |
| Local epochs (FedAvg) | 3 |
| Local batch size (FedAvg) | 10 |
| Epochs (mixture) | 200 with early stopping |
| Batch size (mixture) | 10 |
| Optimizer | Adam |
| Learning rate | 0.0001 |
| Decay rates $(\beta_1, \beta_2)$ | $(0.9, 0.999)$ |
| Training data size per client | $\{5, 10, 20, 38, 50, 62, 75, 88, 100, 500\}$ |
| Majority class fraction $p$ | $\{0.2, 0.3, 0.4, 0.5, 0.6, 0.7, 0.8, 0.9, 1.0\}$ |
| Validation data size per client | 400 |

Table 1: Summary of experiments, including hyper-parameters.

## 4 RESULTS

For the sake of reproducibility, all code will be made available.

In Table 2 we report accuracies and standard deviations on CIFAR-10 for all models when data is highly non-iid, i.e. for $p = \{0.8, 0.9, 1.0\}$. In Figure 3 we report accuracies over all majority fractions $p$. By comparing Figures 3a, 3b and 3c, we see that FEDAVG performs substantially worse when the opt-out fraction increases, i.e. when more clients opt-out from the federation. As a consequence of a weakened global model, the fine-tuned baseline decreases much in performance as well. However, our proposed mixture leverages both the global and local models and does not degrade as much in performance. In Figures 2 and 4 we see that we see that the mixture performs just as well, or better, than the fine-tuned baseline for CIFAR-100 and Fashion-MNIST, respectively.

A common client sampling strategy to mitigate scalability issues in federated learning is to sample a fraction of clients to participate in each communication round of FEDAVG. In Figure 2, we see experiments performed with two different sampling fractions on CIFAR-100. In Figure 2a a client fraction of $1.0$ was used, and in Figure 2b a client fraction of $0.1$ was used. It can be seen that the difference in validation accuracy is low between these two, showing that our proposed method is robust to client sampling.

Experiments were also carried out to see what effect training data size has on performance. This is summarized as a heatmap in Figure 5a for CIFAR-10 and in Figure 5b for Fashion-MNIST, where the difference in validation accuracy between the mixture and the fine-tuned baseline is shown for different majority class fractions $p$ and different client train set sizes. We see here that the mixture model outperforms the baseline most of the time, a trend which gets stronger when the training set size increases.

| $p$ | FEDAVG | Local | Fine-tuned | Mixture |
|-----|--------|-------|------------|---------|
| 1.0 | $27.35 \pm 3.27$ | $84.05 \pm 1.84$ | $85.30 \pm 3.30$ | $\mathbf{86.45 \pm 5.28}$ |
| 0.9 | $35.05 \pm 3.47$ | $75.05 \pm 3.69$ | $75.21 \pm 2.88$ | $\mathbf{75.35 \pm 2.62}$ |
| 0.8 | $39.14 \pm 3.32$ | $65.66 \pm 1.37$ | $65.61 \pm 1.52$ | $\mathbf{67.17 \pm 1.96}$ |

Table 2: Accuracy on unbalanced validation set for highly non-iid majority class fractions $p$ (no class overlap between clients) on CIFAR-10. Accuracies and standard deviations reported are over four runs. Number of clients is 5, opt-out fraction $q = 0.0$ and number of data points per clients is 500.

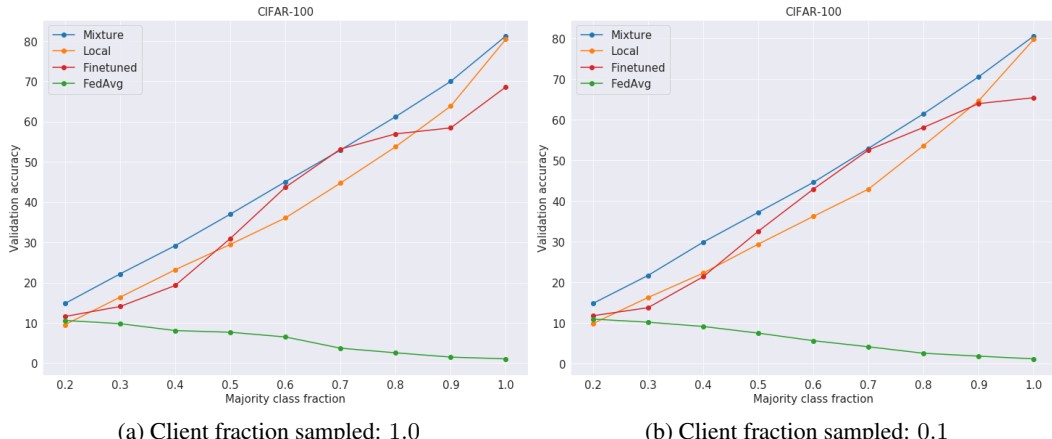

(a) Client fraction sampled: $1.0$        (b) Client fraction sampled: $0.1$

Figure 2: Accuracy on unbalanced local validation data for CIFAR-100 for different majority class fractions $p$. Opt-out factor $q = 0.0$, number of clients is 50 and number of data points per client is 100. Reported values are means over four runs. In (a) we sample a fraction 1.0 of all clients to participate each communication round of FEDAVG, and in (b) we sample a fraction 0.1 of clients.

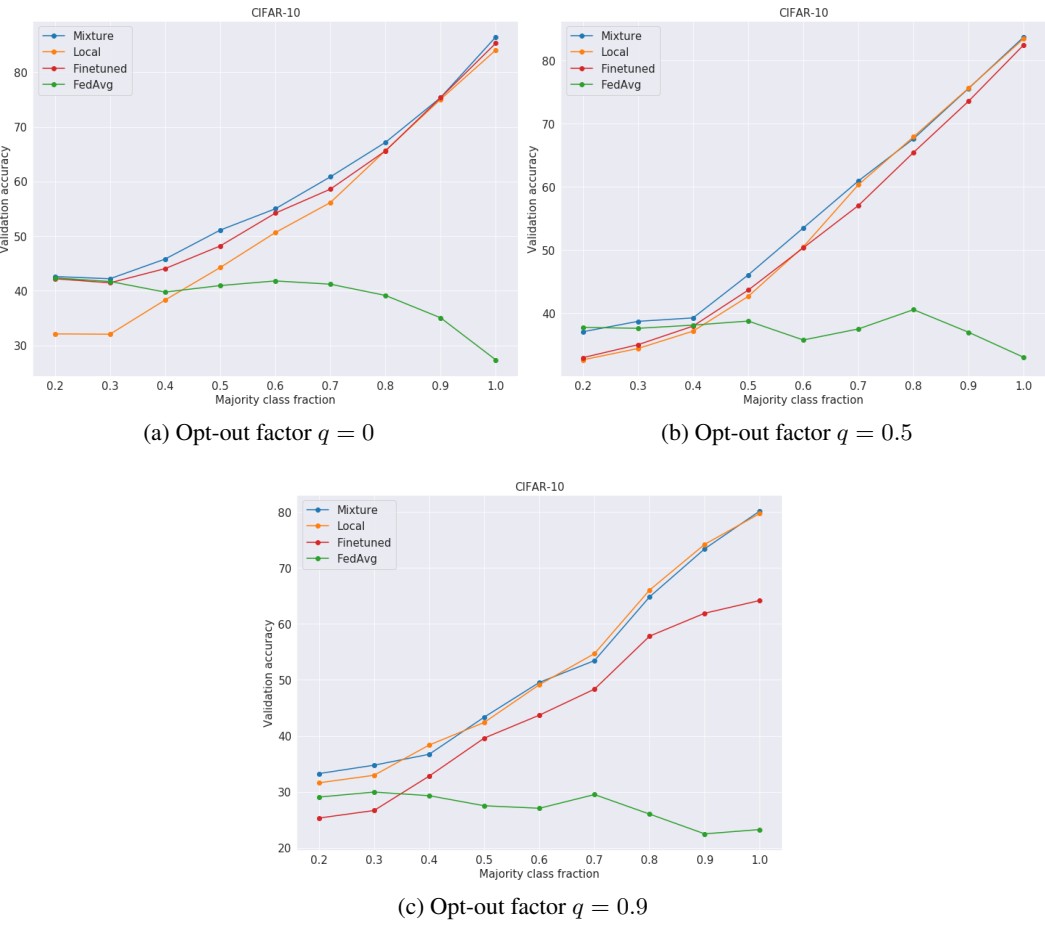

Figure 3: Accuracy on unbalanced local validation data for CIFAR-10 for different majority class fractions $p$. Number of clients is 5 and train set size per client is 500. (a) opt-out factor $q = 0.0$ (b) opt-out factor $q = 0.5$ and (c) opt-out factor $q = 0.9$. Reported values are means over four runs.

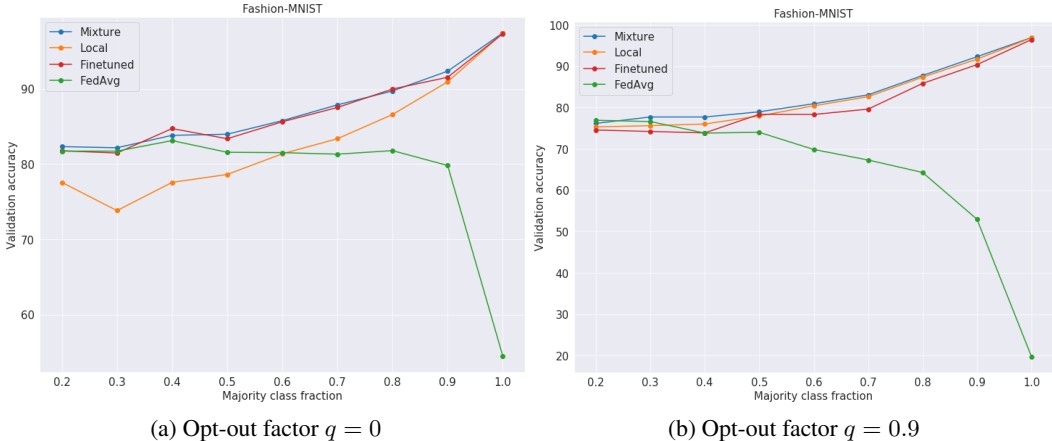

Figure 4: Accuracy on unbalanced local validation data for Fashion-MNIST for different majority class fractions $p$. Number of clients is 5 and number of data points per client is 600. (a) opt-out factor $q = 0.0$ (b) opt-out factor $q = 0.9$. Reported values are means over four runs.

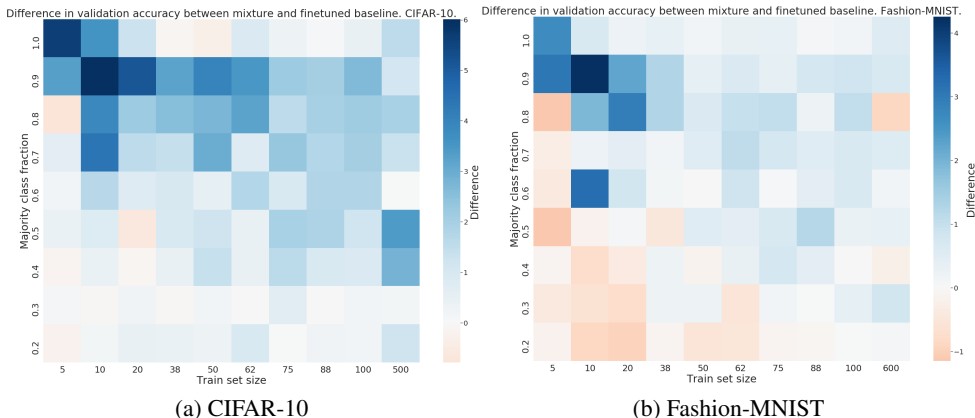

(a) CIFAR-10                                    (b) Fashion-MNIST

Figure 5: Heatmap showing the difference in validation accuracy on CIFAR-10 (a) and Fashion-MNIST (b) for different majority class fractions and client train set sizes between the mixture of experts (ours) and the fine-tuned baseline. Number of clients is 5 and opt-out factor $q = 0.0$. Reported values are means over four runs.

## 5 DISCUSSION

To address the problems of learning a personalized model in a federated setting when the client data is heterogeneous, we have proposed a novel framework for federated mixtures of experts where a global model is combined with local specialist models. We find that with skewed non-iid data on the clients, our approach outperforms all other baselines, including FEDAVG, a locally trained model, and models trained first with FEDAVG and then fine-tuned on each local client in most settings. The experimental evaluation for CIFAR-10 shows that our approach outperforms all other methods, including the strong fine-tuning baseline (see Figure 3). In Figure 5 we see that our proposed model outperforms the fine-tuned baseline in most settings for CIFAR10 and Fashion-MNIST, especially when the number of data points per client increase. For CIFAR-100, the proposed framework outperforms all other methods, regardless of the level of skewness (see Figure 2). In this setting, a large part of the training data for each client comes from a very limited set of the available classes: two out of 100. As such, very few training examples will be available from the minority classes. *This is a crucial result: the proposed framework is very robust to extremely skewed training data.*

The framework also gives strong privacy guarantees, where clients are able to opt-out from federation, keeping their data private. The experiments show that our proposed solution is robust to a high opt-out fraction of users, as seen in Figures 3 and 4, whereas the fine-tuned baseline is not.

## 6 CONCLUSIONS

In this work, we have presented a framework for federated learning that builds on mixtures of experts. This framework allows us to learn a model that balances the generalist nature of the global federated model and the specialist nature of the local models.

Our approach is not only an intuitive approach for the generalist vs specialist balance, but also allows for varying participation of the different clients in the federation. Clients may either opt-in and participate in the federation, or opt-out entirely by training only a local model with all its local data but still receive a global model from the opt-in participants. This gives a flexible solution for strong privacy guarantees in real-world settings. Our experiments show that in the setting where many clients opt-out from federation, the fine-tuned baseline degrades in performance whereas our proposed mixture model does not.

The proposed framework is compatible with any gradient-based machine learning models, and can incorporate combinations of these, strengthening the potential of this direction of research, and leveraging the beneficial properties of ensembles of various machine learning models.

The experimental evaluation conducted in this work showed the proposed solution to achieve state-of-the-art results in three different benchmark datasets when data is highly skewed, and when parts of the clients in the federation opts out from the training.

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
