# OpenReview forum: "Federated learning using mixture of experts"
_ICLR.cc/2021/Conference — Reject_

### Official Review · AnonReviewer4 · 2020-10-26
**what is the motivation of the paper?**

**Rating:** 3
**Confidence:** 4

**Review:**

The proposed method is a federated method allowing to have a certain amount of data shared between all the learners and some data specific to each learner. The targeted field of application is classification for problems where strong privacy is crucial. The method consists in learning a global classifier (with the shared data) as well as local classifiers (one per learner, using the local data). The inference, for each learner, is done with a local expert (another neural network) trained to combine inferences from the local and global models.

The first objection I will make to this paper is the confusion it causes when talking about privacy. Federated learning methods, in a context of privacy protection, aim at building a global model, using private data, without revealing anything about the private data. Here, private data are not used to build the global model, and the local model obtained is not intended to be given, which makes talking about privacy seem irrelevant to me.

For the reason explained above, I wonder about the motivation of the paper. Isn't the best solution, in this case, simply to learn a standard classifier for each local problem, using both local and global data and using a loss function that weights the examples according to the class distributions? It could be argued that if there are many local problems it is more expensive than learning a global classifier and combining it with a local classifier.  In this case efficiency is not privacy is the motivation and the gain should be evaluated in terms of efficiency. This is true, but in this case it becomes a domain adaptation problem: a model in which the distributions of p(y_i) are different from those of the local problem is given and should be used to improve the local problem. In this case, the domain adaptation methods apply and should have been investigated by the authors.

My last comment is on equation (8). I do not understand the relationship between equation (7) and (8). In (7) only the weights of the expert are learned. In (8) all weights are learned. If equation (8) is applied, this means that local data is used to learn the global model through w_g, so there is a leak of local information which contradicts the strict respect of privacy.

---

> ### Author Response · Authors · 2020-11-18
> **Response to R4**
>
> Dear reviewer,
>
> Thank you for your effort and insightful feedback!
>
> We have made some clarifications and corrections in the updated version of the paper. For example, the local models are trained on all available data on a client, both $\mathcal{D}_{\mathcal{I}}$ and $\mathcal{D}_\mathcal{O}$ while the models taking part in the federation are trained only using $\mathcal{D}_\mathcal{I}$. Furthermore, an important point is that no data ever leaves the client. The global model is a federation of the different client models, it benefits from all advantageous properties of federated learning, including certain privacy properties, and the leveraging of data distributed over the clients. Furthermore, the proposed framework is not limited to classification: in fact many different applications can be approached with our method, and a user also has a large freedom in the design of the components (the local models can be any component that can solve the task with some level of success).
>
> I see your confusion about the different levels of privacy in our paper. We have now clarified this in the updated version. Federated learning has been proposed as a means of ensuring some level of privacy, but recent works have shown that an adversary can in some cases recreate the local training data from the weights that are being shared across the network (see e.g. Wang, et.al., 2019). This is why we propose a stronger privacy framework, which at the same time has leverages the strengths of the local models in tackling non-IID data. Therefore, in our framework, each client has the option for two different levels of privacy: the traditional privacy delivered by federated averaging for $\mathcal{D}_\mathcal{I}$ and a stronger privacy for $\mathcal{D}_\mathcal{O}$.
>
> The other central point in our paper is the balance between specialist and generalist. As you suggest, one could approach the specialist end of the spectrum by training only local models weighted by the class imbalances. In our work, we consider settings where there is also a benefit from taking part of the federation. A possible application could be to infer properties from medical texts, such as patient records. A federated set up may be able to leverage text that does not contain any personal information about patients, and at the same time, the model can also leverage the private information with strict siloing.
>
> Since no data ever leaves the clients in our framework, we believe that while domain adaptation is a related problem, it’s solutions aren’t directly applicable to the settings we are considering.
>
> I understand your confusion on equation (8). The writing was not completely clear and there was a typo in eq (8), further eq. (8) is superfluous. In our updated revision, we are removing eq (8) and clarifying eq (7). Further, we are adding an algorithm which explains how we solve the optimization problem.
>
> When the federation is complete (and we have $f_g$) and local training is complete (and we have $f_l^k$), then we optimize the mixture of experts $h^k(x) f_l^k(x) + (1-h^k(x)) f_g(x)$ using local data for each client. In other words, no data is ever leaving a client.

---

### Official Review · AnonReviewer1 · 2020-10-28
**The paper's experiment cannot support its claims.**

**Rating:** 3
**Confidence:** 5

**Review:**


The paper proposes a federated learning framework using a mixture of experts to trade-off the local model and the global model in a federated learning setting. A three-step pipeline is designed to train personalized FL with a mixture of global model and local model.

Pros:

1. The proposed setting is a new scenario that considers both opt-in and opt-out devices.

2. The mixture of the global and local model is a reasonable solution to solve the personalization problem in FL.


Cons:

1. The paper lacks an overall loss function for the whole procedure. In particular, the three steps have three different loss functions, and the updating of the shared global model will cause inconsistent in minimizing multiple loss functions defined by equations 4, 5, and 6.

2. In Section 4, there is a big blank space before the section head and Table 2. The content layout and arrangement could be improved.

3. In Table 1 dataset CIFAR-10, the number of clients is 5 that is very small number for federated setting.  Moreover, the CIFAR-100 data only divided into 50 clients that are also a small number for FL.

4. In Table 2, p = 1 indicates that each client has two classes only. However, the results of FedAvg are 17.13% that is much less than the reported results (85%) of the FedAvg paper by Google.

5. Authors claim the proposed method can protect user privacy since a client can select which data need to be excluded from the federation. However, no corresponding experiments support this claim.

6. Convergence of the gating function is not discussed.

---

> ### Author Response · Authors · 2020-11-17
> **Response to R1**
>
> The authors thank the reviewer for a comprehensive review and for the feedback!
>
> 1. In Equations 4, 5, 6 we use the same loss functions locally. Eq. (4) solves the optimization problem using FedAvg. The losses in eq. (5) and (6) are the same, but for two different models (local model and the mixture model). We will remove the "mix" subscript in eq. (6), as it adds confusion. Overall, we will revise the writing and make this more clear in the paper.  We will also clarify this by adding an algorithm to how we solve the optimization problem.
>
> 2. We will fix the weird layout in the paper for the revision.
>
> 3. We use only 5 clients in CIFAR-10, because we consider the extremely non-iid setting where the data distributions between clients are disjoint. In other words, this is due to no overlap of class labels between clients, which makes the problem harder for FL. This is the same reason as to why CIFAR-100 with a 100 classes is divided into 50 clients. We will clarify this in the experimental section part of the paper.
>
> 4. By the "FedAvg paper by Google", we assume that you mean McMahan et al. 2016. This paper does 1) not use non-iid setting for CIFAR-10,  2) use 5x more data than we per client (we use 100/client) and 3) use 100 clients (allowing label overlap between clients), whereas we use only 5 (in order to fulfill the no label overlap). We will clarify in the paper that we use no label overlap in our experiments. Further, we will add experiments where we use 500 data points per clients as well.
>
> 5. We model this with the opt-out fraction, which decides the fraction of clients which do not participate in the federation. By not participating in the federation, information of the data is never leaving the client which ensures privacy. We have performed experiments on CIFAR-10 for different values of opt-out-fraction. We will make all of this more clear in the revision.
>
> 6. We agree that convergence analysis of the gating function is interesting, and something we want to discuss further in future works.

---

### Official Review · AnonReviewer3 · 2020-10-28
**An interesting combination between Personalized FL and Mixture-of-experts, but need a major revision to improve it.**

**Rating:** 3
**Confidence:** 4

**Review:**

The paper proposed a novel personalized federated learning method using a mixture of global and local models. In particular, a gating function is proposed to leverage the trade-off of two models on the device, and this solution is inspired by a classic work – Mixture of experts (Jacobs, 1991).

Pros:

1. The paper proposes a new federated setting by considering opt-in and opt-out.

2. In step 3, the training of the local mixture-of-experts method doesn’t need to upload data and gradients to the server-side.

3. The proposed personalized federated learning framework is a practical solution.

Cons:

1. The proposed framework requires each client to choose which part of the data is sensitive. This is a very strong assumption in real-world applications.

2. The proposed framework includes three steps. Step 1 & 2 are FedAvg and local supervised learning that are existing methods. Step 3 is to train a personalized local model by mixing local and global models. The mixed-use of global and local models (equation 6) is not a novel way of federated learning. The only novelty part of the method is to apply the mixture-of-experts (Jacobs et al., 1991) method to combine the local and global models.

3. The paper’s writing could be improved. The paper is an integration of the mixture-of-experts method with existing personalized federated learning. The paper's contribution is incremental.

4. The authors should add a discussion to clarify how the old method (mixture-of-experts) can fit into the new environment: federated learning setting and deep learning.

5. To be a paper with self-contained contents, the paper should give a clear definition of the gating model/function h^k, and how to solve the optimization problem described in equation 8.

6. An overall workflow or architecture figure is recommended.

7. An algorithm description is also required.

8. The experiment part is too weak to validate the effectiveness of the proposed method. For example, more datasets and baselines are required.

9. A typo: “We denote the number of clients by k…” It should be “denote the index of clients by k … ”

---

> ### Author Response · Authors · 2020-11-17
> **Response to R3**
>
> The authors thank the reviewer for a comprehensive review and for the feedback!
>
> 1. This may be true. However, in our experiments we model this with some clients opting out from participating in the federation (using the opt-out fraction). This is a very natural real-world setting. By not participating in the federation, information of the data is never leaving the client which ensures privacy. We hade done experiments on CIFAR-10 for different values of opt-out-fraction. We will make all of this more clear in the revision. We will clarify this in the paper revision.
>
> 2. As to our knowledge, using mixture of experts to combine a local and global model to solve the non-iid and personalization problem of federated learning has not been explored prior to this paper.
>
> 3. We agree that the writing could be improved. We will revise this.
>
> 4. We will revise the writing of the paper and make this clearer.
>
> 5. In the revised paper we will clarify that we chose a CNN architecture for h^k in our experiments. We will revise the writing of the optimization section of the paper, and in the revision we will add an algorithm to the paper to address how we solve the optimization.
>
> 6. The overall workflow will be made more clear in the optimization section of the paper, and together with the added algorithm.
>
> 7. This will be added in the revision.
>
> 8. We will add the Fashion-MNIST dataset to the experiments, and discuss it in the revision. As for other baselines, what do you mean? We are comparing with the strong baseline of fine-tuning FedAvg locally.
>
> 9. We will fix the typo.

---

### Official Review · AnonReviewer2 · 2020-10-29
**This work proposes a model personalization method with a gating network that ask fuse the output of the local model and the global model, showing advantages of this approach over fine-tuning the global model with experimental results.**

**Rating:** 6
**Confidence:** 5

**Review:**

1. Strengths

The authors target an important problem in Federated Learning: how to personalize the model to mitigate the nonIIDness.

2. Weakness

The proposed method is not novel. The third step which fine-tunes in the local and global models using a gate network is essentially fusing the global and local models. It is surprising to me that this method works better than fine-tuned after FedAvg. Most importantly, such an empirical method lacks analysis or convincing experimental results.

Hyper-parameters are not well-discussed. The author mention that all experiments use the same learning rate 0.0001. This is definitely misleading. We have to adequately tune the hyper-parameters for each baseline and then make a fair comparison. Using the same learning rate for all baselines are wrong experimental settings. I believe fine-tuning after FedAvg can even get comparable performance if fully tune the hyper-parameters (learning rate, decay, batch size, epochs, rounds, etc).

The authors claim that “client and global models are not constrained to be the same model and could be implemented any two differentiable models.” However, the authors do not provide the experimental result for this argument. I guess when the model architecture is different, the difficulty of hyper-parameter optimization will increase, which weaken the application of the proposed method.

The proposed method has a severe efficiency problem. It requires holding three DNNs at the edge. This is impractical in federated learning where the edge devices are mainly resource-constrained (low memory, low computational ability)

The training time is not mentioned.

The proposed method does not use a client sampling strategy, a common practice in cross-device FL, to mitigate the scalability issue. What the performance if we want to learn 10 thousand sensors? Please check the original FedAvg for details.

The dataset CIFAR10 and CIFAR100 are not difficult enough to demonstrate the concept of the proposed algorithms. Does it still work in a high-resolution setting like ImageNet (224*224). I believe training three DNNs will lead to serious efficiency issues.

The opt-in and opt-out strategy is totally empirical without any intuition about why it works. That the author connects this strategy with a privacy guarantee is somewhat misleading to readers. Please provide an analysis in revision and properly describe the benefit.

In the Introduction section, the following argument is a lack of evidence. Please cite related works to make the argument more convincing.
“Extended phases of local training between communication rounds can similarly break training, indicating that the individual client models will over time diverge towards different local minima in the loss landscape. Similarly, different distributions between client datasets will also lead to divergence of client models”.

The overall writing does not affect my understanding but can be improved.

Related works are not fully discussed. In some knowledge distillation-based method, the personalization is also their benefit. For example, FedGKT [1] also has a personal client model and a global server model, which is just a single step training method without the need of multiple steps training.

[1] Group Knowledge Transfer: Federated Learning of Large CNNs at the Edge. https://arxiv.org/abs/2007.14513

3. Overall Score

Given the above concerns, I recommend reject this paper in the current stage.

4. Questions

May I have the comparison results between "the naive fine-tuning after FedAvg" and the proposed method? Please fully tune hyper-parameters for each baseline.

5. Suggestions

I encourage the authors to do a deeper analysis and a better experimental design. If all the above concerns are addressed, I am happy to increase the score.

---

> ### Author Response · Authors · 2020-11-17
> **Response to R2**
>
> The authors thank the reviewer for a comprehensive review and for the feedback!
>
> 1. The proposed method is not novel.
>
> It is true that combining a local and global model in a federated setting to tackle personalization is not a novel idea, and we discuss this in the related works. However, we have found no other paper using a mixture of experts to solve this task.
>
> 2. Hyper-parameters are not well-discussed.
>
> We have explored different values for lr, and will add that we've done so to the paper. We agree with reviewers that a thorough random search would be useful. We will also add a table summarizing the architectures and hyper-parameters used in the experiments.
>
> 3. The authors claim that “client and global models are not constrained to be the same model and could be implemented any two differentiable models.” However, the authors do not provide the experimental result for this argument.
>
> It is true that we do not show any experiments for other architectures other than CNNs in our experiments. However, our point here is that we propose a framework that is not limited to a certain type of architecture - any differentiable function can be used since our whole model is trained with backprop. We will revise the writing of the paper to make this more clear.
>
> 4. Efficiency problems
>
> The networks we are using are fairly small CNNs. However, we do not aim to solve efficiency problems of federated learning. Our goal is to solve the personalization issue. An interesting area of future research is definitely to solve the efficiency problems of the proposed solution.
>
> 5. Training time
>
> The training time is added in the revised pdf.
>
> 6. Client sampling strategy
>
> It is true that we use 1.0 as a sampling fraction (using all clients) in our federation, as we do not aim to solve efficiency problems of FedAvg. Also, as we only use 5 clients for CIFAR10, sampling fewer clients was not needed. However, we will re-run experiments on CIFAR100 with 50 clients, sampling a subset of clients to participate in the federation to demonstrate that the sampling method works with our proposed solution.
>
> 7. Why not use ImageNet?
>
> We agree with the reviewers remarks that it would be interesting to see if it scales to larger datasets such as ImageNet. However, research withing deep learning based methods in federated learning is relatively new. We have used the most common datasets that researches use within the field of FL (CIFAR10 and CIFAR100). We could not find any paper that has used ImageNet in a FL setting. The problem we aim to solve in this paper is skewness in data (non-iidness). This can happen at any scale, and we hypothesize our model to work also at larger scales - it is definetly a given next step for future research. We will however add experiments on Fashion-MNIST in the revision.
>
> 8. Regarding the opt-in/out strategy
>
> It will be clarified in the paper that in the experiments we run, a client either opts-in (participating in federation) or opt-outs (not participating in federation). However, clients opt-outing will still receive the final global model. This means that the opt-out client's data is completely private, never leaving the client.
>
> 9. the Introduction section
>
> We will add a citation to McMahan et al, 2016 to the introduction in order to back-up the claim of diverging models due to extended phases of local training.
>
> 10. The overall writing
>
> We agree that the overall writing of the paper can be improved, and we will fix this in the revision.
>
> 11. Related works
>
> We will add a citation to the FedGKT paper and discuss it in related works

---

> > ### Comment · AnonReviewer2 · 2020-11-24
> > **Efficient, Scalability, and Privacy are all important aspects of FL**
> >
> > Thanks for your response. It makes me understand your idea much better.
> >
> > But I have to emphasize that in horizontal FL setting, sampling strategy is a must experiment, otherwise we cannot scale to large number of users. Or you can narrow down the applicability to cross-silo setting. This is important because, in practice, if the use number is small, we cannot have enough datasets even like the small scale dataset CIFAR 10.
> >
> > If you can provide experimental results using large scale number of users/clients and demonstrate your method also work well, I am OK to support the acceptence of this paper.

---

> > > ### Author Response · Authors · 2020-11-24
> > > **Sampling technique added to results**
> > >
> > > Thank you for continuing the discussion.
> > >
> > > We have in the results section of the paper added results for CIFAR-100 where we use the sampling technique.
> > >
> > > In Figure 2a a client fraction of 1.0 was used (all clients participate in FedAvg), and in Figure 2b a client fraction of 0.1 (10% random clients participate in FedAvg each round) was used.  It can be seen that the difference in validation accuracy is low between these two, showing that our proposed method is robust to client sampling.

---

> > > > ### Comment · AnonReviewer2 · 2020-11-25
> > > > **I decide to raise my rating since the authors provide sampling-based experimental results**
> > > >
> > > > Thanks. It makes more sense to me.

---

### Decision · Program_Chairs · 2021-01-07
**Final Decision**

**Decision:**

Reject

**Comment:**

The paper presents a personalized federated learning approach using a mixture of global and local models. Four reviewers evaluated this paper; one of the reviewers is luke-warm (6) while the rest of the reviewers pretty negative to this work (3, 3, 3). The reviewers pointed out many weaknesses, especially about novelty, motivation, contribution, presentation, etc. Most importantly, although the idea of a "mixture of experts" makes sense, it is not clear what the real technical contribution of this paper is in terms of federated learning.

Considering all the comments by the reviewers, I believe that this paper is not ready yet for publication. The authors need to improve the novelty and technical soundness of the proposed direction to convince the readers including reviewers.